# Reproducibility and Feasibility of Classification and National Guidelines for Histological Diagnosis of Canine Mammary Gland Tumours: A Multi-Institutional Ring Study

**DOI:** 10.3390/vetsci9070357

**Published:** 2022-07-13

**Authors:** Serenella Papparella, Maria Ines Crescio, Valeria Baldassarre, Barbara Brunetti, Giovanni P. Burrai, Cristiano Cocumelli, Valeria Grieco, Selina Iussich, Lorella Maniscalco, Francesca Mariotti, Francesca Millanta, Orlando Paciello, Roberta Rasotto, Mariarita Romanucci, Alessandra Sfacteria, Valentina Zappulli

**Affiliations:** 1Department of Veterinary Medicine and Animal Production, Unit of Pathology, University of Naples Federico II, 80138 Naples, Italy; papparel@unina.it (S.P.); valeria.baldassarre@unina.it (V.B.); paciello@unina.it (O.P.); 2National Reference Center for the Veterinary and Comparative Oncology (CEROVEC), Experimental Zooprophylactic Institute of Piedmont, Liguria and Valle d’Aosta, 10154 Turin, Italy; mariaines.crescio@izsto.it; 3Department of Veterinary Medical Sciences, University of Bologna, 40126 Bologna, Italy; b.brunetti@unibo.it; 4Department of Veterinary Medicine, University of Sassari, 07100 Sassari, Italy; gburrai@uniss.it; 5Mediterranean Center for Disease Control (MCDC), University of Sassari, 07100 Sassari, Italy; 6Experimental Zooprophylactic Institute of Lazio and Toscana M. Aleandri, 00178 Rome, Italy; cristiano.cocumelli@izslt.it; 7Department of Veterinary Medicine, University of Milan, 26900 Lodi, Italy; valeria.grieco@unimi.it; 8Department of Veterinary Science, University of Turin, 10095 Turin, Italy; selina.iussich@unito.it (S.I.); lorella.maniscalco@unito.it (L.M.); 9School of Bioscience and Veterinary Medicine, University of Camerino, 62032 Camerino, Italy; francesca.mariotti@unicam.it; 10Department of Veterinary Sciences, University of Pisa, 56124 Pisa, Italy; francesca.millanta@unipi.it; 11Independent Researcher, Via Messer Ottonello 1, 37127 Verona, Italy; rr@dwr.co.uk; 12Faculty of Veterinary Medicine, University of Teramo, 64100 Teramo, Italy; mromanucci@unite.it; 13Department of Veterinary Science, University of Messina, 98122 Messina, Italy; alessandra.sfacteria@unime.it; 14Department of Comparative Biomedicine and Food Science, University of Padua, 35020 Padua, Italy

**Keywords:** canine mammary tumours, diagnostic agreement, interobserver variability, classification, standardisation, guidelines

## Abstract

**Simple Summary:**

Tumours of the mammary gland are common in humans, as in canine species. They are very heterogenous with numerous morphological variants and different biologic behaviours. In the last few decades, several efforts have been made to classify these tumours histologically and establish the level of malignancy by using histologic grading systems. However, reproducibility and diagnostic agreement of such classification and grading have been only rarely assessed. In this study, we tested the variability in diagnoses performed by 15 pathologists using the same classification and grading system. Prior to the study, pathologists agreed on guidelines regarding how to apply these systems. Pathologists worked blindly on 36 digital histologic slides of canine mammary tumours. The agreement was statistically analysed using Cohen’s kappa coefficient that, when equal to 1, indicates perfect agreement. The overall agreement in the identification of hyperplastic-dysplastic/benign/malignant lesions was substantial (kappa 0.76), while outcomes on morphological classification had only a moderate agreement (k = 0.54). Tumour grade assigned by pathologists was the least concordant and kappa could not be calculated. Although promising, the results underline that each diagnostic/grading system should be assessed and optimized for standardization and high diagnostic agreement.

**Abstract:**

Histological diagnosis of Canine Mammary Tumours (CMTs) provides the basis for proper treatment and follow-up. Nowadays, its accuracy is poorly understood and variable interpretation of histological criteria leads to a lack of standardisation and impossibility to compare studies. This study aimed to quantify the reproducibility of histological diagnosis and grading in CMTs. A blinded ring test on 36 CMTs was performed by 15 veterinary pathologists with different levels of education, after discussion of critical points on the Davis-Thompson Foundation Classification and providing consensus guidelines. Kappa statistics were used to compare the interobserver variability. The overall concordance rate of diagnostic interpretations of WP on identification of hyperplasia-dysplasia/benign/malignant lesions showed a substantial agreement (average k ranging from 0.66 to 0.82, with a k-combined of 0.76). Instead, outcomes on ICD-O-3.2 morphological code /diagnosis of histotype had only a moderate agreement (average k ranging from 0.44 and 0.64, with a k-combined of 0.54). The results demonstrated that standardised classification and consensus guidelines can produce moderate to substantial agreement; however, further efforts are needed to increase this agreement in distinguishing benign versus malignant lesions and in histological grading.

## 1. Introduction

The histopathological diagnosis and grade of Canine Mammary Tumours (CMTs) are considered the gold standard for patient management and research outcomes [1,2,3]. In this regard, misdiagnosis and/or interinstitutional diagnostic variability between pathologists can seriously affect the interpretation of clinical data that use the histological output as the reference standard. In particular, the evaluation of therapeutic protocols as well as the interpretation of predictive/prognostic molecular markers can be adversely affected [4]. Histopathological diagnostic criteria for CMTs were largely established during years with updated internationally recognized classifications from the World Health Organization (WHO) [5,6] and from the Davis-Thompson DVM Foundation (DTF) [7]. However, whether these classification systems are uniformly applied, as well as the challenges encountered by pathologists in agreeing on the diagnosis and grade of CMTs, is unclear. Two studies were performed on CMT diagnostic agreement already underlying some diagnostic disagreement [8,9]. In addition, significant problems can be related to the application of different classification and grading systems (human versus veterinary; national versus international) and, even when applying common systems, to the subjective interpretation of histological criteria or misleading concepts and terms used to classify and grade CMTs. 

Many factors, therefore, could promote interobserver variability (IOV) and decrease the diagnostic reproducibility among pathologists, resulting in classification and/or grading errors that lead to failure to predict tumour behaviour. Standardisation of diagnosis is also a prerequisite to allow a comparison between research studies worldwide. Efforts for standardisation are common both in human and veterinary medicine and assessment of agreement has been performed on some tumoral and non-tumoral diseases, as non-exhaustively summarised in Table 1 and references therein. Studies evaluating the agreement based on common classification/grading systems or histological criteria underlined that the concordance, despite never being excellent/perfect, is higher when the applied system/criteria are shared and discussed in consensus meetings, thus, establishing the sources of IOV [10,11,12,13,14].

To address the issue of IOV for the diagnosis of CMTs, as part of a national initiative of the Italian Association of Veterinary Pathologists (AIPVET), a group of 15 Italian veterinary pathologists developed national guidelines for CMT assessment using, as a starting point, the last DTF classification of CMTs [7].

Therefore, the aim of this study was to assess IOV in the classification and grading of CMTs when applying the same system and guidelines. The effect of this standardisation on diagnostic concordance and agreement rates was evaluated and critical aspects were pointed out and discussed.

## 2. Materials and Methods

Fifteen veterinary pathologists (see authors list) from academic Schools of Veterinary Medicine, from Veterinary State laboratories (Experimental Zooprophylactic Institutes), and from private veterinary diagnostic laboratories constituted a working group (Working panel, WP) to discuss critical aspects of the recently published DTF classification of CMTs. Two (RR and VZ) of the fifteen pathologists were among the authors in the DTF classification. The WP produced national guidelines [63] and established consensus criteria for histological diagnosis and grading of malignancies for the entities of the DTF classification. For this purpose and due to COVID-19 pandemic, twenty telematic meetings lasting an average of 90 min were held by all components to address common challenges and misconceptions in histopathological assessment of CMTs. Moreover, unified national guidelines were created to reduce the discrepancies between pathologists operating in different institutions and private diagnostic laboratories. The WP identified the principal causes of diagnostic disagreement by interjecting their own direct and indirect (e.g., held seminars and discussions with other colleagues) experiences into the discussion.

More specifically, the consensus regarded the following critical aspects: a. histological subtypes; b. grading; c. criteria for malignancy; d. approach for lymph node metastases and micrometastases; e. pathological prognostic factors; f. markers for phenotype and prognosis; g. content of the histopathological report; and h: application and revision of ICD-O-3.2 codes [64]. ICD-O-3.2 codes have been used now for more than 35 years, principally in human tumour or cancer registries, for coding the site (topography) and the histology (morphology) of the neoplasm, in this way helping standardization. For the purpose of this study, only aspects a) to c) and h) will be presented. 

A consensus on the aforementioned critical aspects was reached and applied during the ring study. The ring study was performed on selected histological samples to evaluate the effect of the classification and the national guidelines in the reproducibility of morphological diagnosis. One experienced pathologist (VZ, ECVP diplomat), internationally recognized for research and continuing veterinary education on mammary gland pathology, selected 34 slides best representing the hyperplastic/dysplastic/neoplastic lesions of the canine mammary gland described in the DTF classification (Table 2). Slides were chosen from the available archive of a university diagnostic veterinary pathology service (BCA Dept., University of Padua, Italy). As per Directive 2010/63/EU of the European Parliament and of the Council of 22 September 2010, regarding the protection of animals used for scientific purposes, the Italian legislature (D. Lgs. n. 26/2014) does not require approval from ethical committees for the use of stored samples in retrospective studies. Additionally, submitting vets sign an informed consent for privacy and to allow the use of protected data regarding samples in research studies. Two more slides were provided by a second participant (RR, ECVP diplomate), also with broad experience on mammary gland pathology. The 36 slides, with minimal repetition of the same histological diagnosis, were progressively numbered (from 1 to 36), digitally scanned (D Sight Menarini) and distributed to the WP for digital examination. Participants were provided the same single hematoxylin-and-eosin-stained slide per case. WP diagnoses were anonymous and blinded to previous and to each other’s interpretations and the diagnosis was recorded providing multiple pre-filled choices of answer to minimize errors (i.e., one drop-down menu for H/B/M and one drop-down menu for the three possible features associated with the scoring for each criterion of the grading). WP participants were asked to interpret the cases following the DTF classification and the newly established national guidelines. They had to classify the lesion as hyperplasia-dysplasia (H) for non-neoplastic lesions, or as benign (B) or malignant (M) for tumours and to identify the specific histological diagnosis also including the corresponding ICD-O-3.2 [64] code as reported in Table 2. Regarding ICD-O codes, the WP analysed the available codes at the time of the study taking as a reference the International Classification for Disease in Oncology ICD-O-3.2. [64] The goal of this action was to update and standardise the cancer codes for the current veterinary cancer registries active in Italy. WP members were also asked to report the histological CMT grading features (i.e., mitotic count, percentage of tubules formation, and degree of pleomorphism) according to Peña and co-authors [2] to calculate the grade. Participants were given a time frame of 4 weeks to complete the evaluations. No clinical details or immunohistochemistry (IHC) results were provided.

### Statistical Analysis

Statistical analysis was carried out by calculating Cohen’s kappa (k) [65,66,67]. The k evaluates the agreement between panellists taking into account agreements due solely to chance. The lacking gold standard was replaced by the mode of the results given by panellists for each sample (majority opinion, GM) [68]. The k is scaled to be 0 when the amount of agreement is what would be expected by chance and 1 when there is perfect agreement between the observers. Kappa values between 0.21 and 0.40 were considered to represent fair agreement; 0.41–0.60 indicated moderate agreement; 0.61–0.80 substantial agreement; and 0.81-1.00 excellent agreement [67]. K was calculated for each panellist versus all (k_ava) and for each panellist versus the GM (k_vGM). The performances of the single panellist were obtained by calculating the mean of the k_ava and of the k_vGM. To synthesise the overall results, a k-combined was calculated for each statistic separately, according to Fleiss and co-authors, indicating the mean of all the k_ava means and k-vGM means, respectively [68]. Statistics were computed for the following parameters: (1) samples identified as H/B/M and (2) specific histological diagnosis, reported as ICD-O-3.2 code (Table 2).

To detect and comment on the differences among the specific histological diagnosis under study, the proportion of cases correctly identified (i.e., cases corresponding to GM) over the total of cases was calculated. This measure does not take into account the effect of chance and, for the purpose of this paper, was referred to as concordance.

The panellists’ experience was then evaluated by performing a hierarchical cluster analysis with the Ward’s method. Cluster analysis was firstly applied to the self-reported variables denoting experience (years of experience, caseload per week, number of published papers), then a second analysis was performed on the classification of lesions as H/B/M. Ward’s method of clustering joins the two groups that result in the minimum increase in the error sum of squares [69].

## 3. Results

### 3.1. Guidelines and WP Composition

For the purpose of this study, the WP discussed some critical points and established and reported into the guidelines a consensus regarding the following aspects.
Histological subtypes—To precisely apply the histological diagnosis reported in the DTF classification, as proposed by the authors. For example, the term “carcinoma in situ” was not applied and instead used atypical hyperplasia or atypical epitheliosis, depending on specific morphological aspects. As another example, it was agreed that the tumour histotype was defined based on the prevalent morphological pattern where more than one pattern was observed (e.g., tubular and solid).Grading—To use the canine grading system proposed by Peña and colleagues [2], as summarised in Table 3. The histological grading was reported, regardless of the presence or absence of vascular invasion.Criteria for malignancy—To employ the following parameter as criteria for malignancy: (I) tumour architecture with reduced tubular organisation (with no objective measurement and no specific cut off); (II) marked cellular and nuclear pleomorphism (with no objective measurement and no specific cut off); and (III) high mitotic count. A cut off ≥6 mitoses per 2.37 mm^2^ was proposed and applied exclusively when other criteria for malignancy were borderline/unclear. This was to indicate the possibility of a lesion with clear evidence of malignancy (e.g., anaplastic carcinoma) and a mitotic count below 6, or of a clearly benign lesion (e.g., ductal adenoma) with a number of mitoses higher or equal to 6. Regarding the mitotic count, it was performed digitally by the WP following these criteria: total area of observation of 2.37 mm^2^ [70] taking into consideration that the digital fields to obtain this total area had to be highly cellular and avoid cystic/necrotic fields. If the expected area (2.37 mm^2^) could not be obtained, the mitotic count was proportionally determined; most mitotically active areas (usually at the periphery of the tumour) were chosen to start, moving to consecutive fields. After two fields with no mitoses, the third new field was chosen as the next new mitotically active field to then proceed again consecutively, and so on until ten counted fields in total. In order to do so, each participant calculated the number of fields to be examined on their screen to cover the standardised 2.37 mm^2^ area. This was done by dividing 2.37 mm^2^ by the total area of a 40“×” U+00D7 image field, which was measured with a ruler tool on the screen [70]. Additional criteria for malignancy were (IV) presence of small areas of random necrosis (groups of neoplastic cells with karyolysis and karyorrhexis), keeping in mind that central wide necrosis can be present both in benign and malignant lesions; (V) peripheral infiltration, determined as an irregular contour of the tumour showing a desmoplastic reaction, often associated with a mixed inflammatory infiltrate; (VI) pluristratification of neoplastic cells with loss of polarity, atypia, and dysplasia; and (VII) lymphatic vessel invasion by neoplastic cells.

WP participants’ characteristics and relative data and experience are shown in Table 4.

### 3.2. Outcomes Expressed in Terms of Hyperplasia-Dysplasia/Benign/Malignant (H, B, M) Showed a Substantial Agreement 

The results, in terms of hyperplasia-dysplasia (H), benign (B), or malignant (M) communicated by individual readers, are reported in Figure 1. 

Only for one case (n.10) it was not possible to establish a GM being equally diagnosed as B/M. Out of 36 cases, 23 cases were 100% concordant (15/15 panellists), among which 17 had a GM = M. Among the 13 discordant cases, only 4 cases were M/B/H discordant. Furthermore, 3 out of 36 cases had 93% concordance (14/15 panellists), 2/3 with a GM = B (14 B and 1M) and 1 with a GM = M (14M and 1 H). The remaining discordant cases were two cases of GM = M versus B discordant, two of GM = B versus M discordant, and two cases GM = H versus B discordant, (Figure 1). The highest discordance was seen for four cases (case 7, 10, 26, 29) with less than 10 panellists agreeing on a diagnosis and having a GM = B or no GM.

Figure 2a shows that the agreement between the participating laboratories was not uniform: the participants, in fact, had an average k ranging from 0.66 to 0.82 (with the 95% CI limits varying between 0.43 and 0.98) and the k-combined is equal to 0.76 (0.74–0.79). The k-vGM, shown in Figure 2b, presents relatively better results than those relating to the k-ava: panellists had average k-vGM ranging between 0.71 and 0.95 (with the 95% CI limits varying between 0.47 and 1.00). The k-combined for the panellists vs. GM was 0.86 (range of means 0.62–1.00).

### 3.3. Outcomes Expressed in Terms of ICD-O Morphological Code/Diagnosis Had a Moderate Agreement

The results expressed by the participants as morphological diagnosis together with the GM are reported in Table 5. As such, 14/15 participants repeated a diagnosis at least once. The estimate of the k for each participant reported in Figure 3a shows that agreement among participants was not uniform: participants had an average kappa ranging between 0.44 and 0.64 (with 95% CI limits ranging between 0.39 and 0.70). The k-combined is equal to 0.54 (95% CI 0.54–0.55). The analysis with respect to the k-vGM, shown in Figure 3b, presents relatively better results than those relating to the k-ava. Panellists had an average k-vGM ranging between 0.52 and 0.94 (with 95% CI limits ranging between 0.47 and 1.00). The k-combined for the panellists vs GM was 0.70 (range of means 0.64–0.76).

A 100% (15/15) or 93% (14/15) concordance among panellists was seen in 6 and 5 cases, respectively, with one additional case (case n. 25) having 2/15 only slightly differing diagnoses (i.e., 13/15 osteosarcoma, 1/15 sarcoma, 1/15 chondrosarcoma). The 5 cases with very high (15/15) concordance included only simple tumours (i.e., composed of one single cell type). They were three cases with GM = M (comedocarcinoma, mucinous carcinoma, anaplastic carcinoma), one with a GM = B (myoepithelioma), and two with GM = H (melanosis of the teat and ductal ectasia). The 5 cases with a high (14/15) concordance had a GM = M in 3 cases ( lipid-rich carcinoma, squamous cell carcinoma, carcinosarcoma), a GM = B in 1 case (ductal adenoma) and a GM = H in 1 case (papillomatosis). 

The most discordant (≥5/15 and <10/15 GM concordant panellists) cases were 11/36 (30.5%). They included six cases with GM = M, among which three had a complex/mixed/solid nature (case no. 24, 30, 31), indicating some difficulties in classifying tumours with a B/M myoepithelial component. Further, one of the six was a ductal-associated neoplasm (GM = ductal carcinoma, case no. 28), one of the six had a spindle appearance (GM = other sarcomas, case no. 20), and one of the six had a combined tubular and papillary pattern (GM = simple tubulopapillary carcinoma, case no. 5). An additional 3/11 discordant cases had a GM = B and included 1/3 cases with a GM = fibroadenoma (case no. 26) mainly differentially diagnosed as hyperplasia with fibrosis, 1/3 case with a GM of simple adenoma (case no. 10), which had one of the lowest concordances (5/15 panellists) and included several differential diagnoses (i.e., ductal adenoma/carcinoma; simple tubular carcinoma; lobular hyperplasia with atypia), indicating a difficulty in identifying also the M/B/H nature. A further 2/11 discordant cases had a GM = H, 1 with a GM = lobular hyperplasia with fibrosis (case no. 18) also diagnosed as fibroadenoma (3/15) or hyperplasia with atypia (4/15) and 1 with a GM = lobular hyperplasia with atypia (case no. 29) associated with several differential diagnoses, including lobular hyperplasia with fibrosis (1/15), simple adenoma (3/15), complex adenoma (4/15), and complex carcinoma (1/15). 

### 3.4. Outcomes Expressed in Terms of Grading 

Since grading was assessed only for samples diagnosed as malignant, a certain amount of heterogeneity was seen. Therefore, we decided not to calculate the k, but to give instead a description of the most discordant elements of grading.

Grading (Appendix A) was never 100% concordant. With regard to those lesions with a GM of a malignant tumour and a GM of histological subtype for which the grading was applicable (15 cases), in 11/15 cases, all the three grades were used by panellists, and in the remaining 4 cases, either grade III or grade I was not applied, two tumours did not reach a GM for grade (n. 30 and n. 36), and the most common GM was grade 2 (9/13). The highest concordance was for one grade II tumour (case n. 5, 73% with 11/15 panellists and with a 100% concordant GM of simple tubulopapillary carcinoma) and for one grade III tumour (case n. 11, 66% with 10/15 panellists and with a 100% concordant GM of comedocarcinoma). Cases with GM = B when diagnosed as malignant (six cases) were predominantly scored grade I, two cases with GM = H diagnosed as malignant were scored as grade I (n.13 by two panellists) and grade II (n. 12 by one panellist). 

### 3.5. Outcomes Expressed Considering Panellist Features 

The cluster analysis performed on the variables synthesising the panellists’ experience pointed out the existence, at the first level of partition, of two groups: one with 11 members and one with 4 members (Figure 4). The four members (3,8,11,12) in the smaller group were identified among the five “experts”. This definition coincides with being considered an expert on CMTs by colleagues, as reported in Table 4. The cluster analysis performed on the classification of lesions, as H/B/M shows, at the first level of partition, found the existence of two groups of eight and seven members. All the experts belong to the first group and cluster among them on the second and the third level of partition (Figure 4). 

The five panellists considered experts by reputation had 100% concordance between them in 28/36 (77.8%) in terms of H/B/M classification, and these cases were always concordant with the GM. With regard to the morphological diagnosis, their concordance was 100% only in 13/36 cases (36.1%) and these were always concordant with the GM. Only in 6/36 (16.6%) cases, their discordance was regarding non-tumoural/benign versus malignant histotypes. 

## 4. Discussion

In this study, we evaluated concordance and agreement in the diagnosis of CMTs applying the same histological classification system (DTF classification [7]), and consensus guidelines [63]. However, as already demonstrated in the literature (see Table 1 and references therein), the overall concordance was below 100% and the overall agreement was below excellent values.

Difficulties in reaching perfect diagnostic consensus are both reader related and lesion related and multiple variables are involved ([71] and references therein).

Considering the reader-related elements, the application of the same classification and grading systems is fundamental for standardization as well as the establishment of international consensus working groups and guidelines [11,12,13,14,71,72]. Nevertheless, even when applying approved systems, high accordance is not easily achieved [71]. In our study, we tested the reproducibility and the feasibility of the DTF classification [7] implemented with national consensus guidelines [63]. No previous studies have been performed on the application of a detailed histological classification and/or guidelines for the diagnosis of CMTs. For human breast cancer (HBC), several attempts have been made (see references in Table 1) obtaining 75% of concordance or a very variable agreement depending on specific subtypes [27,31]. The need for consensus discussions and shared guidelines have, therefore, already been pointed out in human medicine, both for tumoural and non-tumoural lesions, as the classification systems are still too prone to variability of application.

This variability is related to many additional factors. 

Among reader-related factors, the highest is the expertise and the longest is the experience of the pathologists in a specific field, then the highest can be the consensus in that specific area, as demonstrated by our cluster analysis, in which the experienced CMT pathologists are grouped together. In some human studies, a similar higher diagnostic agreement was observed in multivariate analyses as associated with higher diagnostic confidence, similar years of experience, and expertise in a specific area [35,71]. In addition, variable diagnostic approaches can also be pathologist-related aspects, impacting diagnostic variability during the routine, such as the number of sections evaluated per lesion, application of ancillary analyses, such as histochemical and immunohistochemical tests, and a combination of both pathological and clinical aspects to produce a diagnosis [73,74,75,76,77]. All these aspects are very hard to standardise and complex dedicated protocol guidelines should be considered in the attempt of reducing this variability [78,79,80]. They were not targeted in our study but should certainly receive further additional attention.

With regard to lesion-related aspects, some histological features (e.g., cellular/nuclear pleomorphism for establishing malignancy and grading) convey intrinsic qualitative subjective evaluation so that IOV is very hard to minimise [81,82]. Beyond this, biological processes are often a continuum of progressive steps identifying those that necessitate detailed morphological thresholds, which are not always available [83,84]. In CMTs and tumours in general, a major point of discussion is the identification of the transition of a lesion from non-neoplastic to neoplastic and, even more importantly, from benign to malignant [85,86]. In consideration of this point, we investigated the concordance/agreement in the identification of hyperplasia-dysplasia and benign/malignant lesions. Our study showed a relatively good result with the k-combined considered in the literature from “moderate” to “substantial” (means of k_ava ranging from 0.66 to 0.82 with the 95% CI limits varying between 0.43 and 0.98 and the k-combined equal to 0.76) and 23/36 cases with 100% concordance. It can, therefore, be said that the level of histological diagnosis in discriminating between benign, malignant, or hyperplastic-dysplastic lesions was quite satisfactory. However, since k-ava is strongly affected by the diagnosis of each single panellist (i.e., a strong disagreement of only one participant can severely decrease the k), in this study, we also calculated the k-vGM representing the distance of the single panellist from the majority opinion (GM) (ISO13528:2015). With k-vGM, we observed, indeed, an even better agreement (k-combined=0.86, CI 95% 0.62–1.00).

In a similar study conducted in Taiwan, 10 experienced pathologists classified 15 CMTs as either benign or malignant with no further histological classification and, likewise, our study obtained a moderate average level of agreement (0.43k) [9]. Prior to and during the study, these authors did not agree on any specific classification criteria or guidelines; however, they did not include hyperplasia/dysplasia as possible diagnosis, decreasing possibilities of discordance. In our study, a strong discordance was observed in 4/36 cases, in which morphological aspects were overlapping between hyperplastic/benign/malignant lesions. Distinction could be made more on a subjective evaluation than on (missing) objective criteria (e.g., a lesion with a simple tubular organisation with mild atypia can receive a diagnosis of lobular hyperplasia or simple adenoma or simple carcinoma grade I). In the attempt to implement agreement, particularly in these more subjective/borderline lesions, application of specific parameters/thresholds (e.g., mitotic count threshold) were agreed by the WP and probably helped consensus. Application of specific thresholds/methodologies has already been demonstrated to improve concordance in specific areas [49,79]. The parameters applied in our study were taken from the DTF classification and were based on authors’ experience and not on published data. Before establishing precise morphological features/thresholds allowing the identification of tumour progression, the parameters should be carefully evaluated in follow-up large-scale studies, which, however, are very lacking in veterinary medicine [87]. For this reason, the authors still believe that multi-institutional and international application of similar default thresholds would help standardisation, comparison of studies, and collection of large-scale data to assess and possibly redefine the thresholds themselves.

When it comes to the identification of specific histological tumour subtypes, the complexity of the lesions can increase difficulties in reaching blinded consensus diagnoses [7,31,35].

In our study, the agreement on the diagnostic code (identification of a specific histotype) was more unsatisfactory; the average k-combined for k_ava showed values considered in the literature as moderate (0.54k; 95% CI 0.54–0.55). In this case as well, the k_vGM gave a better agreement (k-combined = 0.70, CI 95% 0.64–0.76) suggesting that this type of statistic should always be calculated versus either a standard diagnosis or a majority opinion that is usually lacking within the studies (Table 1). No similar studies have been performed in CMTs. However, similarly to us, two distinct works analysed agreement in classifying canine soft tissue sarcomas and canine and feline nervous system tumours [59] applying specific histological systems and obtained, respectively, moderate (0.60k) and substantial agreement (0.66k) for IOV. In this regard, CMTs are well known for their heterogeneity and complexity of classification [7]. In our study, the tumours characterised by proliferation of myoepithelial cells were included in those lesions receiving less concordance/agreement. The presence of more than one cell type (including myoepithelial cells) in CMTs often requires IHC for definitive characterization; therefore, ancillary tests could be necessary for a definitive diagnosis and should be suggested and accounted for within the report [7]. Additional histostaining and IHC were demonstrated as also improving agreement in other types of tumours [21,59]. 

In our study, tumour grading also showed some discordance, and the agreement could not be calculated because all three grades were frequently applied by WP for the same malignant lesions. Grading has been often found as one of the most reliable prognostic parameters in multivariate analysis [1,2,88]. However, our and other studies, both on humans and on dogs [25] and references therein, indicate that the grading system contains weakly standardizable parameters that can be more easily affected by subjective evaluation [8]. As already reported, a two-tiered system might eventually increase the concordance [10]. In one study evaluating IOV of histological grading of 46 malignant CMTs performed by three mammary pathologists from the same institution, a moderate to substantial agreement (range of kappa means 0.51–0.71) was obtained [8]. This was in accordance with other similar human studies [10,25]. The lowest values were those conferred to nuclear pleomorphism (0.51k) and mitotic count (0.69k) [8]. Evaluation of pleomorphism has already been considered as one of the least concordant features in tumours, due to its heterogeneity within the same tumour and the qualitative subjective nature of the evaluation [10,25]. The mitotic count is instead strongly affected by the selection of areas for the evaluation [8,25,27]. In our study it was performed on digital slides, precisely defining the methodology; however, fields of evaluation varied depending on the starting field that was subjectively established as it was the chosen direction of consecutive fields. Within this framework, digital and computer-aided pathology (CAD), referred to a computational diagnosis system or a set of methodologies that utilises computers or software to interpret pathologic images, are considered emergent fields that will deeply change the temporal and spatial domains of pathologic diagnosis. Thus, CAD systems using machine learning algorithms have been demonstrated to improve classification accuracy and improve reproducibility, reducing the IOV [89,90,91,92].

Taking into account that in veterinary medicine, ring studies to assess IOV are few [8,53,57,59,61] and that a multitude of methodologies are utilised, our study should be interpreted considering some limitations.

First, the pathologists were aware that they were evaluating slides covering nearly all entities present in the DTF classification and this could have influenced interpretive performance, although this bias is likely to have been, at least partially, overcome during the observation of the slides by all the participants who, in the end, repeated the same diagnosis once or twice.

Second, we used only a single section per case. However, in clinical practice, pathologists typically review multiple slides per case and can request additional levels or ancillary immunohistochemical stains to reach a final diagnosis, particularly when more than one cell type is suspected, for example, involving pleomorphic myoepithelial cells [93,94].

Third, being aware of the complexity of CMT diagnoses, the WP carefully defined consensus guidelines based on the DTF classification that could have raised the level of concordance. In order to precisely assess the role of guidelines versus just the DTF classification, a new study should be performed comparing two groups of pathologists applying the same DTF classification and then either using or not the discussed guidelines. The application of guidelines has been already demonstrated as useful in increasing consensus and, therefore, should be considered in addition to or within internationally recognized classification systems [39,71,95].

Further ring studies should be performed, correcting some biases. Surely, the inclusion of more pathologists with even more variable professional expertise form worldwide countries should be considered, in which impact and, therefore, experience in CMTs can be diverse (e.g., Mediterranean countries have more CMTs compared to the US due to cultural attitudes in spaying female dogs) [96,97] and the distribution of cases with variable more realistic frequencies, as it would be in standard routine diagnosis.

## 5. Conclusions

There is no doubt that pathological examination has led to many of the currently used classifications and that morphological observation and its correlation with clinical parameters has provided a sound basis for clinical medicine as it is today. It is also true, however, that subjective histopathological approaches invalidate the overall concepts. Therefore, it is of critical importance to have a diagnosis that is reproducible. The reduction in methodological variables between veterinary pathologists would also improve comparison of studies regarding CMTs. To achieve this goal, we set to revisit the histopathological criteria for diagnosis of CMTs, considering the main findings of all entities described in the last classification of CMTs and to assign a weighting to criteria that drive the diagnosis and grade of these tumours. In this study of pathologists, the overall agreement between the individual pathologists’ interpretation and reference diagnosis (majority opinion) was relatively high when classifying the nature of the lesion (H/B/M), but a bit lower when categorising the specific histotype.

Therefore, several efforts still need to be made to further standardise the application of international classification systems, particularly when approaching heterogeneous diseases, as mammary tumours are in dogs.

## Figures and Tables

**Figure 1 vetsci-09-00357-f001:**
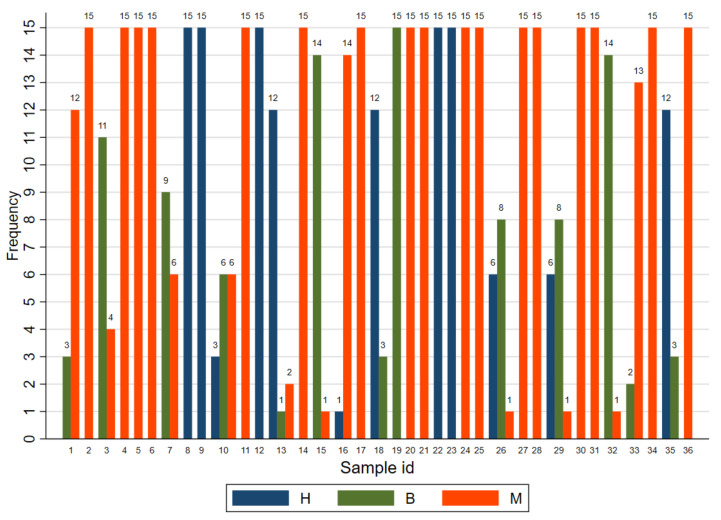
Frequency of results in terms of hyperplasia-dysplasia (H), benign (B), or malignant (M) communicated by individual readers, by sample identification number.

**Figure 2 vetsci-09-00357-f002:**
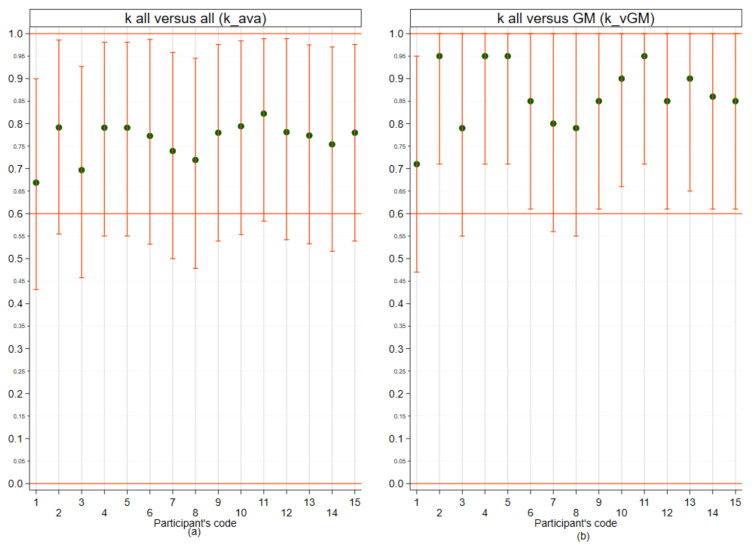
Performances of the panellists for outcomes expressed in terms of Hyperplasia-Dysplasia/Benign/Malignant (H, B, M). (**a**) mean k of each panellist versus all (k_ava); (**b**) mean k of each panellist versus GM (k_vGM). The points (green) represent the mean of Cohen’s k for each panellist; the vertical bars (orange) represent the 95% confidence interval. The horizontal line (red) represents the value above which the agreement is considered substantial.

**Figure 3 vetsci-09-00357-f003:**
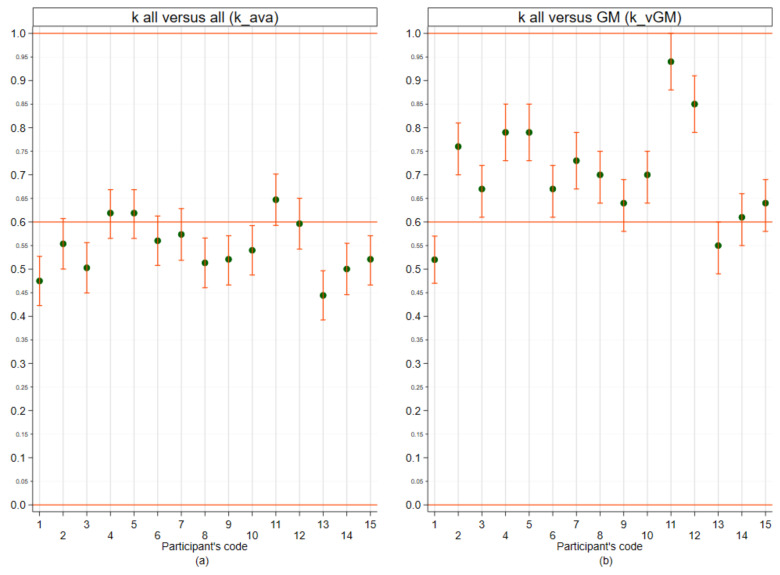
Performances of the panellists for outcomes expressed in terms of ICD-O morphological code/diagnosis. (**a**) mean k of each panellist versus all (k_ava); (**b**) mean k of each panellist versus GM (k_vGM). The points (green) represent the mean of Cohen’s k for each panellist; the vertical bars (orange) represent the 95% confidence interval. The horizontal line (red) represents the value above which the agreement is considered substantial.

**Figure 4 vetsci-09-00357-f004:**
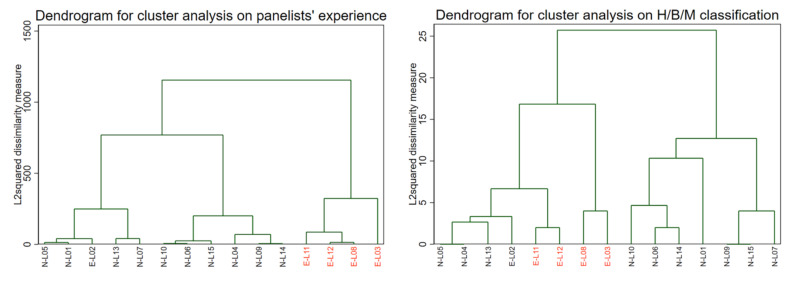
Left, dendrogram resulting after a cluster analysis on the variables synthesising the panellists’ experience; right, dendrogram resulting after a cluster analysis on the classification of lesions as hyperplasia, benign, malignant.

**Table 1 vetsci-09-00357-t001:** Scientific studies dealing with inter- and intraobserver variability in the pathological diagnosis (references are listed in alphabetical order for the topic, separately for humans and animals).

Study	Topic	Species	No. Pat	No. Cases	RD	Lesion Classification	Classification System	Grading System	Outcome	InterO Agreement	IntraO Agreement
Gilles et al., 2008 [15]	brain tumours	H	5	229	NA	histological criteria	NA	NA	weighted k	0.65	NA
Eefting et al., 2009 [16]	cartilagino-us tumours	H	18	16	NA	histological grade	NA	Evans (1977) [17]	weighted k	0.58–0.78	NA
Corazza et al., 2007 [18]	celiac disease	H	6	60	NA	histological criteria	Oberhuber (1999) [19] & Corazza (2005) [20]	NA	weighted k	0.35–0.55	NA
Rugge et al., 2021 [21]	gastric metaplasia	H	3	74	NA	histological criteria & IHC	Nagtegaal (2020) [22]	NA	weighted k	0.7–0.9	0.7 k
Barbosa et al., 2017 [23]	gastric polyps	H	3	128	majority diagnosis	standard diagnosis	Park (2008) [24]	NA	unweighted k	0.40–0.79	NA
Meyer et al., 2005 [25]	HBC	H	7	9000	NA	histological grade & IHC	NA	Elston and Ellis (1991) [26]	unweighted k	0.5–0.59	NA
Longacre et al., 2006 [27]	HBC	H	13	35	NA	standard diagnosis & grade	structured report	Elston and Ellis (1991) [26]	unweighted k	0.3–1	NA
Adams et al., 2009 [10]	HBC	H	5	38	NA	histological grade	NA	Elston and Ellis (1991) [26] vs 2-tiered system	unweighted k	0.32 vs. 0.47	NA
Allison et al., 2014 [28]	HBC	H	3	201	NA	standard diagnosis	NR	NA	% agreement	62.70%	NA
Gomes et al., 2014 [29]	HBC	H	1	610	original report	histological subtypes	Lakhani (2012) [30]	NA	unweighted k	0.22–0.68	NA
Elmore et al., 2015 [31]	HBC	H	115	240	3 panel members	4 categories	NR	NA	% agreement	75.30%	NA
Mäkelä et al., 2018 [32].	lung fibrosis	H	4	60	NA	4 categories	Raghu (2011) [33]	NA	unweighted k	0.4–0.77	NA
Hashisako et al., 2016 [34]	lung (interstitial pneumonia)	H	11	20	NA	histological criteria	Raghu (2011) [33]	NA	unweighted k	0.23	NA
Grilley-Olson et al., 2012 [35]	lung tumours	H	24	96	majority diagnosis	standard diagnosis & categories	Travis (2004) [36]	NA	weighted k, bootstrap for IC	0.25–0.48	NA
Thunnissen et al., 2012 [37]	lung tumours	H	26 (28)	115 (64)	NA	standard diagnosis (invasion)	Yoshizawa (2011) [38]	NA	unweighted k	0.38–0.77 (0.08–0.55)	NA
Nicholson et al., 2018 [39]	lung tumours	H	16	126	NA	histological criteria	Girard (2009) [40]	NA	unweighted k	0.6	NA
Shi et al., 2021 [41]	melanocytic neoplasms	H	3	136	NA	3 categories	NR	NR	unweighted k	0.496	NA
Furness et al., 2003 [42].	renal allografts	H	21	85	NA	histological criteria	Racusen (1999) [43]	NA	unweighted k	0.2–0.4	NA
Ganti et al., 2021 [44]	rhinosinusitis (chronic)	H	2	92	NA	histological criteria	Structured report	NA	unweighted k	0.22–0.64	NA
Hasegawa et al., 2002 [45]	soft tissue sarcomas	H	4	130	expert panel	standard diagnosis, grade & IHC	NR	Hasegawa (2000) [46]	% agreement, unweighted k	75-100%, 0.34–0.86	NA
Denkert et al., 2016 [11]	TILs in HBC	H	32 & 28	120	NA	semiquantitative percentage	web-based & software system	NA	ICC, unweighted k	0.7 & 0.89, 0.45 & 0.63	NA
Tramm et al., 2018 [47]	TILs in HBC	H	9	124	NA	cutoff categories	Salgado (2015) [48]	NA	ICC, unweighted k	0.71, 0.38–0.46	NA
Kilmartin et al., 2021 [49]	TILs in HBC	H	23	49	NA	absolute n. & cutoff categories	scoring digital tool (https://www.tilsinbreastcancer.org)	NA	ICC	0.63 & 0.57	NA
Phytian et al., 2016 [50]	foot lesions	O	8	1158	test standard observer	macroscopic criteria	Hodginkson (2010) [51] & Winter (2004) [52]	NA	unweighted k	0.47–0.72	NA
Lidbury et al., 2017 [53]	liver lesions	C	6	50	NA	scoring system	van den Ingh (2016) [54]	NA	unweighted k	0.16–0.35	NA
Chu et al., 2011 [9]	mammary tumours	C	10	15	NA	benign vs. malignant	NR	NA	unweighted k	0.43	NA
Santos et al., 2015 [8]	mammary tumours	C	3	46	2 panel vet members	histological grade	Goldschmidt (2011) [55]	Karayannopoulou (2005) [56]	weighted & unweighted k	0.5–0.7	NA
Northrup et al., 2005 [57]	mast cell tumours	C	10	60	previous report	histological grade	NA	Patnaik (1984) [58]	% agreement, weighted k	62.1%, 0.62	NA
Belluco et al., 2019 [59]	nervous system tumours	C & F	4	46	neuropathologist	standard diagnosis & IHC	Higgins (2017) [60]	NA	unweighted k	0.66–0.76	NA
Yap et al., 2016 [61]	soft tissue sarcomas	C	3	70	NA	histological criteria & grade	Dennis (2011) [62]	Dennis (2011) [62]	ICC, unweighted k	0.6 & 0.43k	0.78–1 ICC

RD, reference diagnosis; Pat, pathologists; H, human; O, ovine; C, canine; F, feline; HBC, human breast cancer, TILs, tumor infiltrating lymphocytes; NA, not applicable; NR, not reported; k, Fleiss’s/Cohen’s kappa; ICC, intraclass correlation coefficient; IntraO/interO, intraobserver/interobserver.

**Table 2 vetsci-09-00357-t002:** Davis Thomson Foundation classification of canine mammary tumours [7], associated ICD-O-3.2 codes [64], and applied category of lesion. Main categories are indicated in bold and subcategories in italics.

Lesions	ICD-O-3.2 Codes	Category
**1. Hyperplasia/Dysplasia**		
**1.1 Duct ectasia** (DE)	NA	H
**1.2 Lobular hyperplasia** (LH) **(adenosis)**		
*1.2.1 regular (LH-R)*	NA	H
*1.2.2 with secretory activity (LH-S)*	NA	H
*1.2.3 with fibrosis (LH-F)*	NA	H
*1.2.4 with atypia (LH-A)*	NA	H
**1.3 Epitheliosis** (EP)	NA	H
**1.4 Papillomatosis** (PAP)	8060/0	H
**2. Benign epithelial neoplasms**		
**2.1 Simple benign tumours**		
*2.1.1 Adenoma—simple (SAD)*	8211/0	B
*2.1.2 Myoepithelioma (MEP)*	8982/0	B
**2.2 Non-simple benign tumours**		
*2.2.1 Complex adenoma (CAD)*	8983/0	B
*2.2.2 Benign mixed tumour (BMT)*	8940/0	B
*2.2.3 Fibroadenoma (FAD)*	9010/0	B
**2.3 Ductal-associated benign tumours**		
*2.3.1 Ductal adenoma (DAD)*	8147/0 *	B
*2.3.2 Intraductal papillary adenoma (IDPA)*	8503/0	B
**3. Malignant neoplasms**		
*3.1 Carcinoma–in situ*	not applied	
**3.2 Simple carcinomas**		
*3.2.1 Tubular (including cribriform) carcinoma (STC)*	8211/3	M
*3.2.2 Tubulopapillary carcinoma (STPC)*	8263/3	M
*3.2.3 Solid carcinoma (SoC)*	8230/3	M
*3.2.4 Invasive micropapillary carcinoma (IMPC)*	8507/3	M
*3.2.5 Comedocarcinoma (CoC)*	8501/3	M
*3.2.6 Anaplastic carcinoma (AC)*	8021/3	M
**3.3 Non-simple carcinoma**		
*3.3.1 Carcinoma arising in complex adenoma/benign mixed tumour (C in B)*	8941/3 *	M
*3.3.2 Complex carcinoma (CC)*	8983/3	M
*3.3.3 Carcinoma and malignant myoepithelioma (C&MM)*	8562/3	M
*3.3.4 Mixed carcinoma (MC)*	8940/3	M
**3.4 Ductal-associated carcinoma**		
*3.4.1 Ductal carcinoma (DC)*	8147/3 *	M
*3.4.2 Intraductal papillary carcinoma (including papillary-cystic) (IDPC)*	8503/3	M
**4. Malignant epithelial neoplasms-special types**		
**4.1 Squamous cell carcinoma** (SCC)	8070/3 *	M
**4.2 Adenosquamous carcinoma** (ASC)	8560/3 *	M
**4.3 Mucinous carcinoma** (MuC)	8480/3	M
**4.4 Lipid-rich carcinoma** (LRC)	8314/3	M
**4.5 Spindle cell carcinoma** (SPC)	8572/3 *	M
**4.6 Malignant myoepithelioma** (MM)	8982/3 *	M
**5. Malignant mesenchymal neoplasms**		
**5.1 Osteosarcoma** (OC)	9180/3 *	M
**5.2 Chondrosarcoma** (CS)	9220/3 *	M
**5.3 Fibrosarcoma** (FS)	8810/3 *	M
**5.4 Hemangiosarcoma** (HS)	9120/3 *	M
**5.5 Other sarcomas** (other S)	8800/3 *	M
**6. Carcinosarcoma** (CS)	8980/3 *	M
**7. Hyperplasia/dysplasia of the Teat**		
**7.1 Melanosis of the skin of the teat** (Skin M)	ND	H
**7.2 Hyperplasia of the teat** (TH)	ND	H
**8. Neoplasms of the teat**		
**8.1 Benign ductal-associated neoplasms**		
*8.1.1 Ductal adenoma*	8147/0 *	B
*8.1.2 Intraductal papillary adenoma*	8503/0	B
**8.2 Malignant ductal-associated neoplasms**		
*8.2.1 Ductal carcinoma*	8147/3 *	M
*8.2.2 Intraductal papillary carcinoma*	8503/3	M
**8.3 Carcinoma with epidermal infiltration (Paget-like disease)** (C-EI)	8540/3	M

NA, not available; H, hyperplasia/dysplasia; B, Benign tumour; M, malignant tumour. * Code assigned also when name of histotype was different but histological description identical between human and canine lesions.

**Table 3 vetsci-09-00357-t003:** Histological grading for canine mammary tumours [2]. Main categories used for grading are indicated in bold.

Feature	Points
**A. Tubules formation (a)**	
Tubules comprise >75% of the tumour	1
Tubules comprise 10–75% of the tumour (moderate formation of tubules admixed with non-tubular areas)	2
Tubules comprise <10% (minimal or no tubule formation)	3
**B. Nuclear pleomorphism (b)**	
Uniform, regular, small nuclei with occasional small nucleoli	1
Moderate degree of variation in nuclear size and shape, hyperchromatic nucleus, presence of nucleoli (some of which can be prominent)	2
Marked variation in nuclear size, hyperchromatic nucleus, often with more than 1 prominent nucleoli	3
**C. Mitoses per 10 hpf (c)**	
0–9/10 hpf	1
10–19/10 hpf	2
20 or more/10 hpf	3
**Histological malignant grading**	**Totale score (A + B + C)**
I (low, well differentiated)	3–5
II (intermediate, moderately differentiated)	6–7
III (high, poorly differentiated)	8–9

a In complex and mixed tumours, the percentage of tubular formation is scored considering only epithelial areas. In malignant myoepithelioma, tubular formation is 2. In heterogeneous canine mammary carcinomas, tubular scoring should be assessed in the most representative malignant area. b In complex and mixed tumours, nuclear pleomorphism is evaluated in all the malignant components. c HPF, high-power field. The fields are selected at the periphery or the most mitotically active parts of the sample (not only epithelial cells). Diameter of the field of view = 0.55 mm.

**Table 4 vetsci-09-00357-t004:** Characteristics of working panellists at the time of the study.

Age	Sex	Affiliation	Years of Experience *	Position, Titles (In Addition to DVM)	CMTs Biopsies (per Week)	Self-Assessment of Level of Confidence in CMTs (High/Medium/Low)	Considered an Expert by Colleagues on CMTs	Published Papers on MTs°
34	F	Private and University	6	Histopathology Consultant, PhD, ECVP	15	medium	NO	0
46	F	University	15	AP, PhD, ECVP	5	high	YES	14
40	M	University	7	AsP, PhD, ECVP	2	high	YES	8
40	M	IZS°	10	Senior Scientist	10	medium	NO	0
57	F	University	25	PhD	3	medium	NO	2
48	F	University	20	AP, MSc^	4	medium	NO	6
38	F	IZS°	10	Senior Scientist, PhD	7	medium	NO	6
54	F	University	10	AsP	4	medium	NO	4
49	F	University	20	AP	20	high	YES	16
45	M	University	14	FP, PhD	5	medium	NO	0
62	F	University	25	FP, PhD	2	medium	NO	0
38	F	Private	14	Senior Consultant, PhD ECVP	8	high	YES	12
42	F	University	15	AsP	2	medium	NO	1
48	F	University	17	AP, PhD	1	medium	NO	2
46	F	University	15	FP, MSc, PhD, ECVP	6	high	YES	21

* Years interpreting mammary gland pathology cases (not including residency/fellowship training) at 1st June 2021. ^: Master of Science.

**Table 5 vetsci-09-00357-t005:** Classification of histological subtypes by the 15 panellists (P) for the 36 canine mammary tumour samples included in the study. In bold red the diagnoses that differed from the majority opinion (GM), in grey boxes diagnoses repeated by the same panellist.

S-ID	P01	P02	P03	P04	P05	P06	P07	P08	P09	P10	P11	P12	P13	P14	P15	GM
1	IDPA	IDPA	DC	DC	DC	IDPC	DC	IDPC	IDPC	DC	IDPC	IDPA	IDPC	DC	IDPC	
2	IMPC	IMPC	**STC**	IMPC	IMPC	IMPC	IMPC	IMPC	IMPC	IMPC	IMPC	**STC**	IMPC	IMPC	IMPC	IMPC
3	CAD	CAD	**CC**	**CC**	**CC**	CAD	CAD	CAD	CAD	CAD	CAD	CAD	**CC**	CAD	CAD	CAD
4	MC	**C in B**	MC	MC	MC	MC	MC	**C in B**	MC	MC	**C in B**	**C in B**	MC	**C in B**	MC	MC
5	**DC**	STPC	STPC	STPC	STPC	**DC**	**IDPC**	STPC	**IDPC**	**IDPC**	STPC	STPC	STPC	**IDPC**	**IDPC**	STPC
6	**C-EI**	STC	STC	STC	STC	**IC**	**STPC**	**IMPC**	STC	**STPC**	STC	**IC**	STC	**CC**	STC	STC
7	**STPC**	IDPA	**IDPC**	IDPA	IDPA	**STPC**	**IDPC**	**DC**	IDPA	IDPA	IDPA	**IDPC**	IDPA	**DAD**	IDPA	IDPA
8	Skin M	Skin M	Skin M	Skin M	Skin M	Skin M	Skin M	Skin M	Skin M	Skin M	Skin M	Skin M	Skin M	Skin M	Skin M	Skin M
9	DE	DE	DE	DE	DE	DE	DE	DE	DE	DE	DE	DE	DE	DE	DE	DE
10	**STC**	**LH-A**	SAD	**LH-A**	**LH-A**	**STC**	**STC**	SAD	**DC**	SAD	SAD	SAD	**DAD**	**STC**	**DC**	SAD
11	CoC	CoC	CoC	CoC	CoC	CoC	CoC	CoC	CoC	CoC	CoC	CoC	CoC	CoC	CoC	CoC
12	EP	EP	**LH-F**	EP	EP	EP	EP	EP	**LH-R**	EP	EP	EP	EP	EP	**LH-R**	EP
13	**IDPA**	TH	**C-EI**	TH	TH	TH	TH	**C-EI**	TH	TH	TH	TH	**PAP**	TH	TH	TH
14	LRC	**MM**	LRC	LRC	LRC	LRC	LRC	LRC	LRC	LRC	LRC	LRC	LRC	LRC	LRC	LRC
15	DAD	DAD	DAD	DAD	DAD	DAD	DAD	**STC**	DAD	DAD	DAD	DAD	**SAD**	**SAD**	DAD	DAD
16	**TH**	**ASC**	SCC	SCC	SCC	SCC	SCC	SCC	SCC	SCC	SCC	SCC	SCC	SCC	SCC	SCC
17	MuC	MuC	MuC	MuC	MuC	MuC	MuC	MuC	MuC	MuC	MuC	MuC	MuC	MuC	MuC	MuC
18	**FAD**	LH-F	**LH-A**	LH-F	LH-F	**FAD**	V	**LH-A**	**LH-A**	LH-F	LH-F	LH-F	LH-F	**FAD**	**LH-A**	LH-F
19	MEP	MEP	MEP	MEP	MEP	MEP	MEP	MEP	MEP	MEP	MEP	MEP	MEP	MEP	MEP	MEP
20	**SPC**	**SPC**	Other S	**SPC**	**SPC**	Other S	Other S	Other S	**MM**	Other S	Other S	Other S	**FS**	**SoC**	**MM**	Other S
21	AC	AC	**IC**	AC	AC	AC	AC	AC	AC	**IC**	AC	AC	AC	**IC**	AC	AC
22	LH-R	LH-R	LH-R	LH-R	LH-R	LH-R	**LH-S**	**LH-F**	**LH-S**	LH-R	LH-R	LH-R	LH-R	LH-R	**LH-S**	LH-R
23	PAP	PAP	PAP	PAP	PAP	PAP	PAP	PAP	PAP	PAP	PAP	PAP	**LH-S**	PAP	PAP	PAP
24	**C in B**	MC	MC	**C in B**	**C in B**	**C in B**	**C in B**	MC	MC	**C in B**	MC	MC	**C in B**	MC	MC	MC
25	**ChS**	OS	OS	OS	OS	**ChS**	OS	OS	OS	OS	OS	OS	**Other S**	OS	OS	OS
26	**LH-F**	FAD	FAD	FAD	FAD	**LH-F**	FAD	FAD	**LH-F**	**LH-A**	FAD	FAD	**C&MM**	**LH-A**	**LH-F**	FAD
27	**SoC**	LRC	LRC	LRC	LRC	LRC	LRC	**SoC**	LRC	**SoC**	LRC	LRC	**Other S**	LRC	LRC	LRC
28	**SoC**	DC	**MM**	**SoC**	**SoC**	**SoC**	DC	**MM**	DC	**STC**	DC	DC	DC	DC	DC	DC
29	**SAD**	**SAD**	**CAD**	**CAD**	**CAD**	**SAD**	LH-A	LH-A	LH-A	CC	LH-A	LH-A	**CAD**	**LH-F**	LH-A	LH-A
30	**IDPC**	**IDPC**	**SoC**	**IDPC**	**IDPC**	C&MM	C&MM	C&MM	**SoC**	**ASC**	C&MM	C&MM	**SoC**	**STPC**	**SoC**	C&MM
31	**MM**	C&MM	C&MM	C&MM	C&MM	**MM**	**CC**	**SPC**	**CC**	C&MM	**CC**	**CC**	**MM**	C&MM	**CC**	C&MM
32	BMT	BMT	**C in B**	BMT	BMT	BMT	BMT	BMT	**CAD**	BMT	BMT	BMT	BMT	BMT	**CAD**	BMT
33	ASC	**IDPC**	ASC	ASC	ASC	ASC	**IDPA**	ASC	ASC	**C-EI**	ASC	ASC	ASC	**IDPA**	ASC	ASC
34	CS	CS	CS	CS	CS	CS	CS	CS	CS	CS	CS	CS	**OS**	CS	CS	CS
35	LH-S	LH-S	LH-S	LH-S	LH-S	LH-S	**SAD**	LH-S	**SAD**	LH-S	LH-S	LH-S	**LH-A**	LH-S	**SAD**	LH-S
36	SCC	SoC	AC	HS	HS	HS	SoC	SoC	AC	HS	SoC	SoC	HS	AC	AC	

S-ID, sample identification; see Table 2 for other abbreviations.

## Data Availability

Original data of samples are within institutional archives and have mainly been submitted through the platform www.simbavet.org, however personal data on clinicians, owners and animals are not available to the public.

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
