# Peer review of "Reproducibility and Feasibility of Classification and National Guidelines for Histological Diagnosis of Canine Mammary Gland Tumours: A Multi-Institutional Ring Study"

_vetsci, 2022, doi:10.3390/vetsci9070357_

Round 1
Reviewer 1 Report
This manuscript “Reproducibility and feasibility of classification and national guidelines for histological diagnosis of canine mammary gland tumors: a multi-institutional ring study” reports the intra-observer study of 36 CMT cases diagnosed by 15 veterinary pathologists. The authors underlined the histological assessment of CMT had high reproducibility. But the agreement of grading of CMT was low between pathologists.
- Had authors tested the inter-observer agreement of tubules formation, nuclear pleomorphism or mitotic score? Replaced the criteria with low agreement by a quantified scale in an attempt to reduce subjectivity.
- Authors showed the cluster analysis of the panellists’ characteristics. Have any feature of the panelist correlated with H/B/M diagnosis or histology grading?
Author Response
vetsci-1733298
Reviewer 1
This manuscript “Reproducibility and feasibility of classification and national guidelines for histological diagnosis of canine mammary gland tumors: a multi-institutional ring study” reports the intra-observer study of 36 CMT cases diagnosed by 15 veterinary pathologists. The authors underlined the histological assessment of CMT had high reproducibility. But the agreement of grading of CMT was low between pathologists.
- Had authors tested the inter-observer agreement of tubules formation, nuclear pleomorphism or mitotic score? Replaced the criteria with low agreement by a quantified scale in an attempt to reduce subjectivity.
As grading was expressed only for samples diagnosed as malignant, it was assessed sparsely among samples. Moreover, as grading was not independent from the H/B/M agreement, we felt that calculating an inter-observer agreement on grading could be not only imprecise, but misleading. Therefore, we preferred to describe results without reporting measures.
2. Authors showed the cluster analysis of the panellists’ characteristics. Have any feature of the panelist correlated with H/B/M diagnosis or histology grading?
Conceptually, hierarchical agglomerative linkage clustering proceeds as follows. The N observations start out as N separate groups, each of size one. The two closest observations are merged into one group, producing N - 1 total groups. The closest two groups are then merged so that there are N -2 total groups. This process continues until all the observations are merged into one large group, producing a hierarchy of groupings from one group to N groups. The panellists’ features are considered as a whole in this process, therefore it is not possible to distinguish the single feature that characterizes the dissimilarity. We hope this can explain better and answer to the comment.
thank you again
Reviewer 2 Report
In this study, the authors provided an extensive quantification of the reproducibility and an analysis of the interobserver variability of histological diagnosis and grading of canine mammary tumors. Overall, they observed a moderate to substantial agreement of histological classification. This is an important study that aims to highlight the relevance of a standardization and a nationwide or, rather, worldwide consensus on the histological diagnosis of tumors, particularly when they have such a high heterogeneity, such as canine mammary tumors. The study is novel and relevant to the field. The analysis is well done. There are a few minor points that need to be addressed before the manuscript can be considered suitable for publication.
Table 1 has too many columns. As a result, it looks very busy. I would suggest deleting the column “location of pathologists”, as I believe it does not add a significant value to the table. Also, please merge columns “study” and “year” and report only the citation (number). Similarly, also column “classification system” could report only the citation. Finally, consider using abbreviations in the column “species” (H for human, C for canine, and so on).
Line 124. Can the authors provide some additional details regarding their strategy of “multiple pre-filled choices (…) to minimize errors”?
Line 141. Is the citation #65 placed in the correct position within the sentence? I would move it to the end of the sentence unless there is a specific reason why it is there.
The authors stated that the term “carcinoma in situ” was not applied. Can the authors provide an explanation for that? I wonder why this entity is in the DTF classification considering it is rarely applied, or not applied whatsoever, like in this study.
Line 192-194. The sentence “When 2-3 consecutive fields ….. and so on.” is not clear. Can the authors provide an explanation and perhaps rephrase the sentence? In other words, it is not clear whether the fields without mitoses were counted or not.
Table 4 contains sensitive information (i.e., age). Based on authors’ affiliations (title page) and on geographical locations and other information present in this table, it would be easy to know the age of each author, which is considered sensitive information. I wonder if the authors are aware of that. Also, I believe that the geographical location does not add any value to this table. I would cancel the columns “age” and “geographical location”. However, if the authors would like to keep the column “age” because it provides a reference on how experienced each author is, I would at least then cancel the column “geographical location”.
Table 4, column “considered an expert…”: please modify “YES” and “no” such that they match (either UPPERCASE or lowercase).
Lines 215-216. The authors stated that it was not possible to establish a GM for cases 10 and 29. While this is true for case 10, I believe it is not true for case 29, as 6 panelists reported H and 8 reported B. Isn’t in this case GM = B?
I am not sure whether including in the WP also the author who selected these CMT cases would be fair. This will inevitably skew the analysis towards a “perfect agreement”. Looking at figures 2 and 3 as well as table 5, I wonder if panelist 11, who always shows the highest k, is the author who selected the cases….
Table 5 is a bit busy. Please adjust the column width so that all the columns have the same width. Also, using acronyms (tubular carcinoma ïƒ STC, tubulopapillary carcinoma ïƒ STPC, and so on) for each tumor diagnosis would perhaps help having the table in 1 page. Also, can the authors clarify what the grey boxes refer to? The legend is not clear.
Table 5. Looks like GM of sample #1 is missing.
In the analysis, the authors decided to calculate k versus GM. In some cases, the GM was different from the diagnosis reported by the “experts”. As an example, case #4 has a GM = mixed carcinoma, whereas the “experts” (panelists 8, 11, 12) reported carcinoma arising in CAD/BMT. Now, why did the authors give more value to the diagnosis reported by the majority, rather than to the diagnosis reported by the experts? I believe this does not affect the analysis highlighted in figure 2, but it does affect the analysis shown in figure 3 (ICD-O codes). Could the authors please comment on this? Would it make sense to add a “k all versus experts” analysis of ICD-O codes?
I would probably emphasize a bit more on the importance of immunohistochemistry to make a diagnosis for those lesions that have more than one tumor population.
The discussion section is very long. Would it be possible to make it a bit shorter?
A few sentences throughout the manuscript are a bit hard to follow, mainly because they are too wordy. As an example, please consider cutting the sentence in the abstract that describes the aims of the study (lines 31-35).
Please modify “panellists” in “panelists” throughout the whole manuscript.
Line 289: please modify “more” in “most”.
Line 310: please modify “clusterize” in “cluster”.
Author Response
In this study, the authors provided an extensive quantification of the reproducibility and an analysis of the interobserver variability of histological diagnosis and grading of canine mammary tumors. Overall, they observed a moderate to substantial agreement of histological classification. This is an important study that aims to highlight the relevance of a standardization and a nationwide or, rather, worldwide consensus on the histological diagnosis of tumors, particularly when they have such a high heterogeneity, such as canine mammary tumors. The study is novel and relevant to the field. The analysis is well done. There are a few minor points that need to be addressed before the manuscript can be considered suitable for publication.
Table 1 has too many columns. As a result, it looks very busy. I would suggest deleting the column “location of pathologists”, as I believe it does not add a significant value to the table. Also, please merge columns “study” and “year” and report only the citation (number). Similarly, also column “classification system” could report only the citation. Finally, consider using abbreviations in the column “species” (H for human, C for canine, and so on).
Thank you very much for these suggestions that make the table less complex. If possible, we would prefer to keep just the name of authors, years and applied systems since some of them are known and it would be easier for the
readers to have the information. Also, having the year of the publication directly into the table would give a direct idea of how old/new are the studies
Line 124. Can the authors provide some additional details regarding their strategy of “multiple pre-filled choices (…) to minimize errors”?
Thank you for the comment, we included this explanation into the text: (i.e., one drop-down menu for H/B/M and one drop-down menu for the three possible features associated with the scoring for each criterion of the grading, see table 3).
Line 141. Is the citation #65 placed in the correct position within the sentence? I would move it to the end of the sentence unless there is a specific reason why it is there.
Thank you, this was a mistake and the ref was moved to the end of the sentence
The authors stated that the term “carcinoma in situ” was not applied. Can the authors provide an explanation for that? I wonder why this entity is in the DTF classification considering it is rarely applied, or not applied whatsoever, like in this study.
Thank you for this comment. Authors know that this is a topic under debate. The DTF proposes not to apply the diagnosis of in situ k until when diagnostic criteria will be better standardized. Because of this and since no additional staining or IHC could be performed during this study to better define an in situ lesion, authors preferred to use the more general and wide inclusive diagnoses of atypical hyperplastic/dysplastic lesions.
Line 192-194. The sentence “When 2-3 consecutive fields ….. and so on.” is not clear. Can the authors provide an explanation and perhaps rephrase the sentence? In other words, it is not clear whether the fields without mitoses were counted or not.
We are sorry that this was not completely clear. We rephrased into the text as: After 2 fields with no mitoses, the third new field was chosen as the next new mitotically active field, to then proceed again consecutively, and so on until 10 counted fields total.
Table 4 contains sensitive information (i.e., age). Based on authors’ affiliations (title page) and on geographical locations and other information present in this table, it would be easy to know the age of each author, which is considered sensitive information. I wonder if the authors are aware of that. Also, I believe that the geographical location does not add any value to this table. I would cancel the columns “age” and “geographical location”. However, if the authors would like to keep the column “age” because it provides a reference on how experienced each author is, I would at least then cancel the column “geographical location”.
Thank you for noticing this. Authors are aware and accepted the information included into table 4. We anyway canceled the column with geographical locations
Table 4, column “considered an expert…”: please modify “YES” and “no” such that they match (either UPPERCASE or lowercase).
Modified, sorry for the inattention .
Lines 215-216. The authors stated that it was not possible to establish a GM for cases 10 and 29. While this is true for case 10, I believe it is not true for case 29, as 6 panelists reported H and 8 reported B. Isn’t in this case GM = B?
Modified, sorry for the inattention . It Is true, case 29 had a GM of B
I am not sure whether including in the WP also the author who selected these CMT cases would be fair. This will inevitably skew the analysis towards a “perfect agreement”. Looking at figures 2 and 3 as well as table 5, I wonder if panelist 11, who always shows the highest k, is the author who selected the cases….
Cohen’s k was calculated either by comparing each pair of panellists or comparing each panellist with the GM. Therefore, even if the author who selected the cases was included in the study the analysis would not be skewed versus perfect agreement: k=1 is reached only if there is a perfect agreement between 2 panelists or between a panelist and the GM (a mode between 15 panellists). The performance of a single panellist, therefore, depends on the agreement with each of the other panellists. Panelist 11 was an expert and his/her performance did not significantly differ from other experts
Table 5 is a bit busy. Please adjust the column width so that all the columns have the same width. Also, using acronyms (tubular carcinoma ïƒ STC, tubulopapillary carcinoma ïƒ STPC, and so on) for each tumor diagnosis would perhaps help having the table in 1 page. Also, can the authors clarify what the grey boxes refer to? The legend is not clear.
Yes, authors are aware that this was a very busy table. Thank you for the suggestions, we tried to make the suggested changes. The grey boxes indicate the diagnoses repeated by the same panelist, we rephrased it into the table title.
Table 5. Looks like GM of sample #1 is missing.
Yes this is missing since 6 IDPC and 6 DC were diagnosed
In the analysis, the authors decided to calculate k versus GM. In some cases, the GM was different from the diagnosis reported by the “experts”. As an example, case #4 has a GM = mixed carcinoma, whereas the “experts” (panelists 8, 11, 12) reported carcinoma arising in CAD/BMT. Now, why did the authors give more value to the diagnosis reported by the majority, rather than to the diagnosis reported by the experts? I believe this does not affect the analysis highlighted in figure 2, but it does affect the analysis shown in figure 3 (ICD-O codes). Could the authors please comment on this? Would it make sense to add a “k all versus experts” analysis of ICD-O codes?
The experts did not fully agree on all the diagnosis (for example for case #4 expert 3 gave mixed carcinoma), therefore we should have calculated firstly the GM among experts, then compared it with each non expert rater (losing the 5 expert raters, i.e. 41% of panellists). Using all panellists would have overestimated the k, because the 41% of panellists were compared with their own restricted GM). Using the GM among all panellists according to us was the better option. Surely we already planned a new ring test with more pathologists and more random distributed cases selected from a colleague which will not be a panelist.
I would probably emphasize a bit more on the importance of immunohistochemistry to make a diagnosis for those lesions that have more than one tumor population.
On order not to extend the discussion further, we just modified a sentence into the discussion as: The presence of more than one cell type (including myoepithelial cells) in CMTs often require IHC for definitive characterization, therefore, ancillary tests could be necessary for a definitive diagnosis and should be suggested and accounted for within the report [7]. We hope this can be enough.
The discussion section is very long. Would it be possible to make it a bit shorter?
We tried to short down and simplify some sentences.
A few sentences throughout the manuscript are a bit hard to follow, mainly because they are too wordy. As an example, please consider cutting the sentence in the abstract that describes the aims of the study (lines 31-35).
The sentence of the abstract was shortened and rephrased, and we tried to simplify some sentences. thank you
Please modify “panellists” in “panelists” throughout the whole manuscript.
Pannellist with two “L” should be UK English and this was the language selected for the manuscript. We will leave the choice to the editor. Thank you
Line 289: please modify “more” in “most”
Modified
Line 310: please modify “clusterize” in “cluster”.
Modified
Reviewer 3 Report
What an interesting article, particularly for anyone working in pathology. The reproducibility of diagnoses is a major concern for those who work in the field and also for clinicians, who need to trust who signs the paper.The work is well designed, and the authors are aware of its limitations. Interestingly, even with pathologists from the same background (presumably), the results can be as different as the same lesion is seen as malignant or benign. It is a path that the pathology still has to follow, defining less subjective criteria for the classification of CMT. Thank you for your contribute.
Author Response
Thank you very much indeed! we hope we can produce more studies like this one and standardize better our way to make diagnoses.
Reviewer 4 Report
The manuscript "Reproducibility and feasibility of classification and national guidelines for histological diagnosis of canine mammary gland tumours: a multi-institutional ring study" is interesting and addresses a classic pathological problem associated with interobserver variability, but little studied in canine mammary tumors. The text is well written and is clear in its objectives and results. There are some minor observations that are detailed below:
Introduction
- line 120-121: define ICD-O-3.2 code
Results
- check the wording in line 242
- Table 5 is not shown. Please add
- In my opinion, Figure showing the cluster analysis must be in the manuscript and not as a supplementary Figure
Author Response
thank you very much.
We included a description of ICDO codes
Regarding line 242 in our version this corresponds to: Out of 36 cases, 23 cases were 100% concordant (15/15 panellists) among which 17 had a GM = M. Among the 13 discordant cases only 4 cases were M/B/H discordant. Furthermore, 3 out of 36 cases had 93% concordance (14/15 panellists), 2/3 with a GM = B (14 B and 1M) and 1 with a GM = M (14M and 1 H). The remaining discordant cases were 2 cases of GM = M versus B discordant, 2 of GM = B versus M discordant, and 2 cases GM = H versus B discordant, (Figure 1).
Sorry but we are not sure what you would like us to rephrase in this paragraph/lines.
- Table 5 is not shown. Included - we need the editorial office to adjust the layout
- In my opinion, Figure showing the cluster analysis must be in the manuscript and not as a supplementary Figure. Included - we need the editorial office to approve to add the figure.